# Influence of Body Image, Risk of Eating Disorder, Psychological Characteristics, and Mood-Anxious Symptoms on Overweight and Obesity in Chilean Youth

**DOI:** 10.3390/bs14030154

**Published:** 2024-02-21

**Authors:** Paula Lizana-Calderón, Jesús M. Alvarado, Claudia Cruzat-Mandich, Fernanda Díaz-Castrillón, Mauricio Soto-Núñez

**Affiliations:** 1Department of Psychobiology and Behavioral Sciences Methods, Faculty of Psychology, Complutense University of Madrid, Campus Somosaguas, Carretera De Húmera, s/n, 28006 Madrid, Spain; jmalvara@ucm.es; 2Centro de Estudios de la Conducta Alimentaria (CECA), Faculty of Psychology, Universidad Adolfo Ibáñez, Av. Diagonal Las Torres, 2640, Santiago 7941169, Chile; claudia.cruzat@uai.cl (C.C.-M.); fernanda.diaz@uai.cl (F.D.-C.); mauricio.soto@edu.uai.cl (M.S.-N.)

**Keywords:** overweight, obesity in young people, path analysis, body image, eating disorders, body mass index (BMI)

## Abstract

This study investigates the relationship between body image, eating disorders, psychological characteristics, and mood and anxiety symptoms in Chilean youth, with nutritional status, particularly overweight and obesity. With a sample of 1001 participants from five regions of Chile, aged 15 to 23 years. The Eating Disorder Inventory 3 (EDI-3), the Multidimensional Body-Self Relations Questionnaire Appearance Scales (MBSRQ_AS), and the Symptom Inventory Derogatis Revised (SCL90-R) and a sociodemographic questionnaire were used to analyze these variables. A model including nine exogenous (independent) variables and ten endogenous variables, based on a literature review, was evaluated by path analysis. The results show a significant association between factors such as sex, family history of overweight, self-classification by weight, and body dissatisfaction with body mass index (BMI). Eating behaviors such as overeating, and lack of appetite were also found to be influenced by interpersonal sensitivity, overweight preoccupation, and drive for thinness. The study underscores the importance of promoting a positive body image and addressing overweight/obesity from a combined health psychology and public health perspective, highlighting the need for interventions that consider nutritional status, and in particular overweight and obesity, as a phenomenon with multifactorial causes and maintainers.

## 1. Introduction

Obesity is the most prevalent food-related clinical condition worldwide. It is considered a chronic disease and a public health problem, with progressively greater incidence [1,2,3]. According to the World Health Organization report, the global age-standardized prevalence of obesity in adults has increased 1.5-fold from 2000 to 2016. In 2016, 39% of adults worldwide were overweight and 13% were obese [4].

### 1.1. Obesity in Chile

In 2016, Chile had the highest rates of overweight and obesity among OECD countries, with 40% of the adult population being overweight and 34% being obese [5].

To address this problem, various public health intervention programs and initiatives have been developed in Chile to reduce obesity rates, using prevention strategies mainly focused on nutrition and amount of exercise, such as: “Elige vivir sano”, “El plato de tu vida”, “Kioskos saludables”, “Vida Sana”, implementing Law #20,606, and others [6,7]. Unfortunately, the latest National Health Survey indicated that despite these efforts, obesity continues to increase nationwide [8]. This was associated with the fact that the recovery of people with obesity is difficult, largely because of the high dropout rates observed in prevention and treatment programs [9]. For example, the “Vida Sana” program implemented by the Chilean Ministry of Health experienced an 86% dropout rate before 12 months [10].

Thus, despite the efforts made, the prevalence of overweight and obesity in Chile has not improved and remains at concerning levels [5]. This highlights the need for a deeper understanding of this disease.

Recent evidence, such as the narrative review by Sánchez-Carracedo [11], proposes that obesity should be considered as a chronic, progressive, complex, prevalent, and recurrent disease, resulting from the interaction between environmental, behavioral, metabolic, and genetic factors. Although obesity is widely recognized as a multifactorial problem [12,13,14,15,16], it is still often viewed as primarily a physical health issue. This perpetuates the misconception that obesity is solely the result of a lack of self-discipline and personal responsibility [11]. The limited success of country-level programs can be attributed to their focus on combating unhealthy habits, such as abnormal eating behavior and increased sedentary lifestyles, without fully understanding the complexity of the phenomenon [17].

### 1.2. Mental Health and Obesity

Several research studies have investigated the factors associated with the development and maintenance of overweight and obesity. Among them, some studies [18,19] have highlighted a strong relationship between mental health indicators and obesity. They identified that the key appreciation of body image, self-esteem, and symptoms associated with stress, anxiety, and depression [20] (e.g., emotional dysregulation [21]) are psychological factors strongly linked with the clinical condition and which are interrelated. The importance of establishing these links is that effective interventions to reduce obesity rates require mental health spaces that provide tools for emotional regulation, psychological well-being, and promote body satisfaction [18,19,21,22].

In turn, various studies [23,24,25,26] have confirmed the relevance of psychological factors relevance in body image dissatisfaction among overweight or obese individuals compared to those with a normal nutritional status. This is correlated with a greater concern about their figure. These results are consistent with the findings of Kamody et al. [27], who have identified that people who are overweight or obese, together with high importance on their body shape, tend to experience a greater weight gain and worse perceived physical and mental health. However, perceiving oneself as having a “normal” or “thin” body is considered a protective factor against depressive anxiety symptoms and future weight gain. The explanation of this phenomenon is that individuals in the former group tend to focus on their physical appearance, neglecting a holistic view of themselves that takes into account other aspects beyond their figure. This narrow focus can lead to a negative perception of their health, increased anxiety symptoms, and lower levels of self-esteem [27].

### 1.3. Psychosocial Factors and Obesity

In addition to the psychological factors mentioned above, studies have shown that there are significant gender differences in the assessment of obesity. Women tend to experience higher levels of body dissatisfaction than men and are at greater risk of developing eating disorders (ED) [19,28,29]. This situation places females at a higher risk of developing problems with their general mental health. These problems may include the appearance of anxious and depressive symptoms, as well as the emergence of disordered and/or pathological eating attitudes that may have originated during adolescence [28,30].

Simultaneously, other studies [31,32,33] have investigated the relationship between obesity and psychosocial factors. Notably, these studies have highlighted the concepts of sensitivity and interpersonal distrust.

Interpersonal sensitivity refers to feelings of inadequacy and inferiority towards others, which can impede the development of meaningful relationships. Calugi and Dalle [31] assessed the relationship between obesity and psychosocial variables. They concluded that interpersonal sensitivity is linked to internalized weight stigma, general psychological distress, and lower desired body mass index (BMI). Therefore, it can be considered a determinant psychosocial factor in those seeking obesity treatment.

Interpersonal distrust refers to the difficulty in expressing personal thoughts and emotions, due to feelings of disappointment, disillusionment, and alienation in social relationships. In this regard, Calderón et al. [32] state that as body weight increases, concern for the social environment also increases along with insecurity in relationships. This can lead to the rise of social problems and tendencies towards self-isolation. La Marra et al. [33] explain that interpersonal mistrust is sustained by the idealization of the culture of thinness, which generates prejudices and negative attributions towards those who are overweight or obese. This discrimination hinders the psychological well-being of these individuals. The results of this preliminary study suggest the coexistence of two seemingly contradictory trends, which may in fact be two aspects of the same issue. Obese adolescents expressed widespread dissatisfaction with their body image, coupled with a desire to lose weight. Conversely, they also showed a tendency towards depersonalization and limited self-awareness specifically related to the most dissatisfactory body parts.

On the other hand, Haire-Jushu et al. [34] argue that obesity should be considered as an intergenerational disease. This is because both parents can transmit biological aspects and/or behavioral patterns that can promote or avoid the risk of obesity in their children and generate epigenetic modifications.

The factors that play a relevant role in obesity can be visualized. Obesity is linked to the development, maintenance, and/or intensification of symptoms associated with various health disorders, including ED and somatic symptom disorder (SSD).

### 1.4. Relationship between Obesity, Eating Disorder, and Somatic Sympton Disorder

Bray et al. [35] and Balantekin et al. [36] conducted studies on the relationship between obesity and ED. They found that both pathologies share risk and maintenance factors related to mental health indicators. To understand this, it is necessary to consider that EDs encompass a range of behaviors and attitudes that arise from a constant and intense preoccupation with appearance and weight. This preoccupation leads to body dissatisfaction which can trigger risky behaviors and disordered eating, such as lack of appetite or excessive food intake. These behaviors can be further exacerbated when the person presents obesity [35,36].

Along the same lines, Kim et al. [37] link obesity with somatic symptom disorder, identifying that people with SSD and high BMI tend to experience greater severity in their somatic symptoms. However, it is important to note that this positive association does not arise from a direct influence of BMI on the intensity of somatization. The relationship between obesity and SSD is mediated by dysfunction in working memory. This is due the fact that obesity can cause systemic inflammation, cognitive distortions, and impacts the brain regions associated with working memory, which can aggravate somatic symptoms [37]. It is important to note that somatization disorder is one of the most common mental health disorders, along with depression and anxiety disorders [38]. The prevalence of somatization disorder highlights the urgency to develop effective treatments for obesity.

In summary, obesity is a significant and increasing public health concern, necessitating the development of more effective strategies to address it. In order to enhance the efficacy of prevention and treatment policies, it is crucial to gain a comprehensive understanding of the phenomenon by integrating the factors described above, which have been studied separately despite their potential simultaneous occurrence.

The research aims to propose an initial model that allows for understanding overweight and obesity in young Chileans, from a multifactorial perspective. This will enable the development of interventions at both public health and clinical levels, with better results than those achieved so far.

The present study addresses the following research question: what is the influence of body image assessment, risk of eating disorders, certain psychological characteristics, and mood-anxiety symptoms on overweight and obesity in Chilean youth?

The study hypotheses were as follows. It was expected that high mood-anxiety symptoms, difficulties in social relationships, lower self-esteem, body dissatisfaction, and risk of eating disorders would be linked to BMI in young people. This is based on the idea that emotional and psychological tensions can influence eating habits and nutritional status [22,31]. Secondly, the study expected that satisfaction with appearance and low concern about overweight would be inversely related to BMI in young people. The hypothesis proposed that a positive body image and less concern about weight gain could help maintain a healthy nutritional status [27]. Additionally, it was expected that there would be a greater presence of symptomatic severity and general psychological maladjustment as BMI increased. The hypothesis was that young people with weight problems may experience psychological difficulties, which could have significant implications for their emotional and mental well-being [37,39].

## 2. Materials and Methods

### 2.1. Participants

Initially, a sample of 1346 students was selected by non-probabilistic quota sampling. A total of 345 cases were excluded due to systematic missing data on weight or height that did not permit BMI calculation, missing values in the responses to the instruments, or subjects who did not meet the inclusion or exclusion criteria (e.g., age ≥ 15 years and <24). The eliminated cases were compared with the remaining cases with no significant differences found in terms of the proportion of men and women (χ^2^(1) = 0.105, *p* = 0.746). The distributions of the responses corresponding to the model variables between the complete and discarded cases were analyzed using the Kolmogorov–Smirnov test for independent groups, finding significant differences only in age (Z of K-S = 4.026, *p* < 0.001). Regarding age, a difference of 0.894 years was found in the mean age in favor of the final sample, which is not relevant.

Consequently, the sample consisted of 1001 Chilean adolescents and young people, with 431 males (43.1%) and 570 females (56.9%). Their ages ranged from 15 to 23 years, with an average age of 18.93 years (SD = 2.30); 46.8% were between 15 and 18, while the rest were 19 or older. The young people came from the Metropolitan Region, one coastal region (V region), and three regions in the center-south of the country (VI, VII, and VIII regions).

The inclusion criteria were to be an adolescent or young person between 15 and 23 years old, of both sexes, students, and residing in the regions indicated. Exclusion criteria: having been under nutritional, psychological, or psychiatric treatment with a diagnosis of eating disorder.

The sample size meets the recommendations of Kline [40] suggesting a minimum of between 10 and 20 cases for each parameter to be estimated.

To recruit participants, various schools, and universities in five regions of the country were contacted. Authorization was obtained from the institutions, and the participants provided their informed consent by signing the corresponding signature. Minors were required to sign an informed consent form, while their parents or legal guardians also signed an informed consent form. This consent was approved by the Bioethics Committee of the National Commission for Scientific and Technological Research of Chile, CONICYT, and students and parents kept a copy when applicable. All questionnaires were anonymous. Participation in the study was voluntary and no financial benefit was provided to participants. The procedure was supervised and approved by the Ethics Committee at Universidad Adolfo Ibáñez.

### 2.2. Instruments

First, participants completed a sociodemographic data questionnaire, as shown in Table 1:

To evaluate the variables included in the model, the following instruments were applied:

SCL-90-R, Derogatis Symptom Inventory-Revised [41], whose functioning was evaluated in Chilean university students [42]. (It should be noted that there is limited research on the psychometric properties of the instrument in Latin American adolescent populations [43,44], and specifically in Chile, there is a lack of studies. However, the SCL 90-R is an instrument that can be applied to adolescents in the general population [45], and its utility in adolescents and young adults is emphasized [46]). The SCL-90R consists of 90 items, grouped into nine primary dimensions or scales: Somatization, Obsessions, Interpersonal Sensitivity, Depression, Anxiety, Hostility, Phobic Anxiety, Paranoid Ideas, and Psychoticism. Participants are asked to indicate how often they have experienced each symptom on a five-point Likert scale (from “never” to “very often”). It includes three global index scales: Symptom Severity Index (GSI), Psychological Difficulties Index (PSDI), and Positive Distress Index (PSDI-POS). These scales provide an overview of symptom severity and overall individual distress levels.

Multidimensional Body-Self Relationship Questionnaire Appearance Scales (MBSRQ_AS) [47]. An abbreviated self-administered questionnaire with 34 items, whose original version in Spanish was validated by Botella et al. [48] and whose psychometric properties were analyzed in a Chilean sample [49]. Unlike the more extensive questionnaire (MBSRQ), this version focuses on perception and attitudes toward physical appearance, including five body image dimensions: Appearance Evaluation (AE); Appearance Orientation (AO); Overweight Preoccupation (OP); Self-classification by Weight (SCW); and Body Area Satisfaction (BAS). The dimensions are explained in more detail in Table 2. Items from 1 to 22 are scored using a 5-point Likert scale (1 = strongly disagree to 5 = strongly agree). In the Body Area Satisfaction (BAS) scale, different parts of the body are listed, assigning scores between 1 (very dissatisfied) and 5 (very satisfied). In the Self-Classification by Weight (SCW) scale, respondents are asked to classify their weight from their personal perspective and from others’ opinion, and it is composed of two items that score between 1: very underweight and 5: very overweight. Finally, item 23, referring to the attempt to lose weight quickly through extreme dieting, should be answered between 1: never and 5: very often.

Eating Disorder Inventory 3 (EDI 3) [50] is an updated and expanded version of the EDI 2, which is composed of 3 ED risk scales, Drive for Thinness (DT); Bulimia (B) and Body Dissatisfaction (BD), and 9 scales that assess psychological aspects associated with the development and maintenance of ED: Low Self-Esteem (LSE); Personal Alienation (PA); Interpersonal Insecurity (II); Interpersonal Alienation (IA); Interoceptive Deficits (ID); Emotional Dysregulation (EDy); Perfectionism (P); Asceticism (A); Maturity Fear (MF). It also allows the calculation of six additional indices (EDs risk, inefficacy, interpersonal problems, affective problems, excess control, and general psychological maladjustment). The psychometric properties of the EDI 3 were recently evaluated in a young Chilean population [51].

The subscales and items of each instrument identified in the model appear in the following table:behavsci-14-00154-t002_Table 2Table 2Summary of subscales in the instruments used.MBSRQ_ASSCL90-REDI 3Appearance Evaluation (AE): Satisfaction with body shape and physical attractiveness.Somatization (SOM): discomfort due to perception of bodily symptoms dependent on the autonomic nervous system.Drive for Thinness (DT): thinness obsession.Appearance orientation (AO): preoccupation with physical appearance. Includes what others perceive.Interpersonal sensitivity (IS): feelings of inferiority and inadequacy.Low self-esteem (LSE): basic concept of negative self-evaluation, implying insecurity, ineffectiveness, maladjustment, and perceived inability to achieve personal goals.Overweight preoccupation (OP): concern about weight or about doing things to maintain or change weight if necessary.Depression (DEP): clinical manifestations of depressive symptoms.Interpersonal alienation (IA): feelings of disillusionment, disappointment, alienation, and distrust in relationships.Self-classification by weight (SCW): individual weight evaluation from the perspective of oneself and others.Anxiety (ANS): nervousness, tension, and panic attacks.Emotional dysregulation (EDy): tendency to emotional instability, impulsivity, and intolerance to certain emotional states.Body Areas Satisfaction (BAS): degree of satisfaction with parts and aspects of my body.Item 19: lack of appetite. Item 60: overeating.



### 2.3. Statistical Analysis

The distributions of all the variables to be included in the model were evaluated using the Kolmogorov– Smirnov test (Lilliefords), maintaining the null hypothesis of univariate normality only in Low Self-Esteem, Interpersonal Alienation, and Appearance Evaluation. Multivariate normality was tested using the Mardia’s test, obtaining skewness = 3421.6, *p* < 0.001 and kurtosis 23.3, *p* < 0.001; the null hypothesis was thus rejected.

For the variables’ distributions, we decided to evaluate the fit of their relationship by Path analysis (SEM) with MLR estimator (maximum likelihood estimation with robust standard errors), which is robust for non-normal continuous variables [52]. The analyses were performed using Mplus 8.5 software [53].

To calculate the subscale scores of the EDI-3 and MBSRQ_AS instruments, the factor scores were used according to the analyses previously performed [49,51]. For the SCL90-R, we followed the indications of the authors of the Chilean validation [42].

The hypothesized model first considered the dimensions of body image that had shown a direct relationship with BMI in previous studies (SCW, OP, AO), as well as interpersonal relationships (IA) and emotional aspects (Edy). It aimed to identify relationships between gender, age, and low self-esteem with these variables. Subsequently, depressive, and anxious symptoms related to risk behaviors (Overeating and Lack of Appetite) were included in the model, organizing the remaining variables in interrelation, based on substantive criteria, either as antecedents or covariates, obtaining a model composed of 19 variables, 9 of which are exogenous.

The evaluation of model fit was based on χ^2^, Comparative Fit Index (CFI); Tucker-Lewis Index (TLI); Root Mean Square Error of Approximation (RMSEA); and Standardized Root Mean Square Residual (SRMR), using the criteria indicating that values above 0.95 for CFI and TLI would account for an optimal fit and greater than 0.90 would be acceptable; for RMSEA, values under 0.06 would be considered optimal and below 0.08, acceptable. For SRMR, the criterion is values lower than 0.06 [54].

## 3. Results

For the definition of the model to be evaluated, we first identified those variables that, according to the literature review, could have a direct influence on the nutritional status of young people (BMI). First, the direct effects of all the predictor variables on BMI were evaluated, and finally sex, family history of overweight, three dimensions of body image from the MBSRQ_AS (Overweight Preoccupation, Self-classification Weight, Appearance Orientation), Interpersonal Alienation, and Emotional Dysregulation were maintained. Subsequently, concerning the aforementioned variables, we included as predictors those that constitute a risk for developing an ED (Obsession with Thinness, Overeating, and Lack of Appetite) and finally characteristics more related to aspects of mood (Depression) and Anxiety, Somatization, and Self-evaluation about the appearance and areas of the body, together with the ability to achieve personal goals. Interpersonal Sensitivity was also taken into account, as it could influence anxiety levels and account for aspects of body image about interaction with others (e.g., Appearance Orientation). Age was incorporated in correlation with Body Mass Index, since the index increases with age, especially during the adolescent and early adult age range of the sample. As per Pérez et al. [55], the model was re-specified based on the parameters’ modification indexes.

The final model presents a good fit, with CFI = 0.968, TLI = 0.954, SMRM = 0.033, RMSEA = 0.036 (see Table 3), consisting of nine exogenous (independent) variables and ten endogenous variables. Of these, nutritional status (BMI), Low Self-Esteem, and the behaviors of Overeating and Lack of Appetite are exclusively dependent variables, as shown in Figure 1, where only the statistically significant coefficients are presented (*p* < 0.05).

For the Age variable, a significant coefficient of 0.113 was detected about self-classification by weight (SCW), which was considered spurious because it had no theoretical support and because this relationship could be due to the correlation of age with BMI (r = 0.201) (see Table A1 Appendix A).

## 4. Discussion

Regarding the predictor variables of nutritional status in young Chileans (refer to Table 4), the direct effect of Sex (−0.621) associated with body build is noteworthy, with males having a higher BMI. The family history of being overweight is a very relevant variable (0.441), which may be due to genetic factors or the perception of overweight or obesity as an intergenerational pathology, i.e., behavioral patterns of relationship with food would also be learned within families, in line with Haire-Joshu and Tabak [34].

Of particular interest, is the total effect obtained for self-classification by weight (SCW) (0.637). This suggests that the self-perception of overweight or obese status could account for a tendency to experience greater weight gain, as observed by Kamody et al. [27], constituting a risk factor.

In relation to the variables stated in the first and third hypotheses, only minor direct effect of Emotional Dysregulation (0.063) on BMI was observed. This is despite the fact that EDy has previously been linked to overeating behavior [21]. The same minor effect on BMI was observed for Interpersonal Alienation (0.049). However, the predictor effect of the other variables is mediated by other variables, as shown in Figure 1.

However, the body image dimensions (Appearance Evaluation, Overweight Preoccupation) exhibit relationships that are consistent with those reported in the literature (−0.193; 0.127, respectively). This means as BMI increases, the evaluation of appearance worsens and there is a greater concern about weight [23,24,25], which supports the study’s second hypothesis. Additionally, Appearance Orientation, which includes aspects of others’ perception, decreases with higher BMI (−0.121). The relationship between Appearance Orientation and Interpersonal Alienation (−0.107), suggest that people in alienated interpersonal relationships tend to avoid appearance concerns because as they interfere with such relationships. This phenomenon is supported by the findings of La Marra et al. [33], which indicate that depersonalization may decrease attention to one’s own body. Further research is warranted on this aspect, taking into account variables that were not included in the present study, such as teasing or weight stigma, or evaluating whether the relationship between both factors is nonlinear.

In addition, prior research [29] has found that women tend to place a greater emphasis on their appearance (0.245) and to exhibit a higher self-classification by weight (0.309), which is a risk factor for both ED [36] and mental health problems [35].

In relation to Emotional Dysregulation, the coefficient obtained is low (0.063). This variable is influenced by anxious (0.211) and depressive (−0.122) symptomatology. Higher levels of anxiety are associated with increase emotional dysregulation, while greater depressive symptomatology appears to decrease it, possibly due to the low levels of activation of depressive symptoms [15]. Casagrande et al. [22] suggest that overweight individuals may have more difficulty recognizing emotions (alexithymia) and experience greater emotional dysregulation compared to those with normal weight. This emotional dysregulation could potentially pose a risk factor linked to higher BMI and a poorer prognosis for overweight and obesity issues. On the other hand, an increased BMI, may lead to emotional dysregulation due to efforts to self-regulate, along with the social and personal consequences it entails.

In terms of personal evaluation, particularly regarding the variables associated with to Low Self-Esteem (inefficacy), the influence of family History of Overweight (−0.224) is noteworthy as it reduces personal insecurity. This finding is similar to Kamody et al. [27] regarding the transmission of eating patterns within families, which may involve self-evaluation in terms of family body aspects. Additionally, it was observed that self-perceived inefficacy is inversely related to Appearance Evaluation (−0.242 meaning the negative assessments of personal capabilities increase when the latter worsens. The study found that Body Areas Satisfaction (−0.144) had a weaker negative correlation with the phenomenon, which is consistent with previous research [18,19]. The coefficients for the risk factors for ED, Overweight Preoccupation and Drive for Thinness, were −0.122 and −0.055, respectively, indicating that a greater sense of inefficacy is associated with less concern for weight and thinness, although DT has a lower magnitude. This perspective would be reasonable, given that such concern is interpreted as a proactive willingness to take action to maintain or adjust weight if needed or a set of inflexible belief aimed at achieving thinness. Finally, the coefficient obtained for Somatization (−0.132), contradicts the expected findings. However, according to Kim et al., [37], the association of this symptomatology with nutritional status would be mediated by cognitive functions that were not included in the present investigation.

Finally, when analyzing the risk behaviors of ED, Overeating, and Lack of Appetite, it is evident that the former is directly and strongly influenced by Interpersonal Sensitivity (0.305) and Overweight preoccupation (0.153) and inversely by Drive for Thinness (−0.246), which aligns with Calugi and Dalle [31]. Regarding Lack of Appetite, the direct influence of depressive symptomatology is highlighted (0.309) as was also noted by Chow and Tan [19]; Somatization is noteworthy (0.239), which according to Kim et al. [37], could also be a factor contributing to a higher prevalence of malnutrition. The indirect impact of Interpersonal Sensitivity, Drive for Thinness, and Overweight Preoccupation on Lack of Appetite can be explained by their inverse relationship with Overeating (−0.339).

The limitations of this study predominantly are mainly due to the self-reported nature of BMI measurements, which may compromise data accuracy. As noted by Allison et al. [56], BMI and weight were likely underreported, while height was overreported in both sexes, particularly among obese individuals. This could be related to the low proportion of self-reported participants in the “obesity” category in the sample, compared to national prevalence indicators. However, some research suggests that self-reporting may not accurately predict BMI at an individual level. Despite this, it remains a simple and valid method for estimating BMI among overweight and obese individuals in epidemiological studies [57].

The study had limitations as it was unable to assess variables related to the nutritional status, body image, and mental health of adolescents, such as physical activity and social media use [58,59]. Future research should consider including these factors and external evaluation of BMI, to improve accuracy and validity of the results.

Another significant limitation of the study is that the objective was to evaluate a model that included a large number of predictor variables to account for the multiple influences on the nutritional status of young people. This ultimately complicates the interpretation. Therefore, it is recommended to view this initial model as an exploratory approach and continue in the direction of simplifying it to aid comprehension and application in designing of specific interventions.

## 5. Conclusions

According to the main objective of this research, the model evaluated allows us to affirm that there is a significant relationship between body image, eating disorders, psychological characteristics, and mood symptoms among Chilean adolescents and young adults, especially those who are overweight and obese.

Among the findings, factors such as gender, family history of overweight or obesity, self-perception of weight, and body dissatisfaction are associated with BMI. In addition, eating behaviors such as overeating and loss of appetite appear to be influenced by Interpersonal Sensitivity, Overweight Preoccupation, and Drive for Thinness.

Women tend to experience higher levels of Appearance Orientation and Self-Weight Classification than men and are at greater risk for eating disorders, placing them at risk for problems in their overall mental health.

The study shows the importance of promoting a positive body image as a protective factor against overweight and/or obesity, the relevance of considering familial nutritional status characteristics and possibly their eating patterns, and the need to design interventions against overweight/obesity from a multidimensional perspective, considering, for example, factors such as emotional dysregulation, anxiety, and depressive symptomatology.

In summary, obesity is a serious and growing public health problem in Chile, which needs to be addressed by more effective strategies. This study provides a deeper understanding of the phenomenon, integrating several factors that have been studied separately, and suggests that all of them can be presented, analyzed, and considered simultaneously. Furthermore, the complexity of the proposed model makes it necessary to replicate it in other samples in order to more precisely determine the relevance and stability of the observed relationships.

## Figures and Tables

**Figure 1 behavsci-14-00154-f001:**
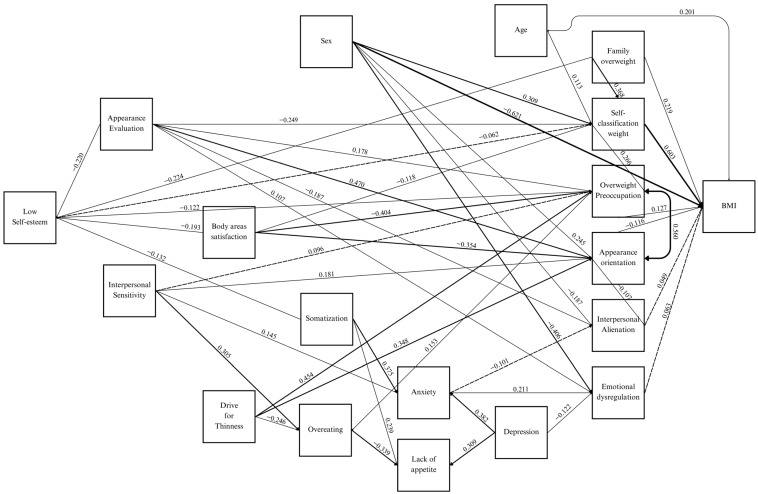
Path analysis model of nutritional status in Chilean youth, according to body image, risk of eating disorders, psychological characteristics, and mood-anxiety symptoms.

**Table 1 behavsci-14-00154-t001:** Sociodemographic characteristics.

Sociodemographic Characteristics	N	%
Total	1001	100
Sex		
Male	431	43.06
Female	592	56.94
Age (years)		
15–17	265	26.47
18–20	446	44.56
21–23	290	28.97
Nutritional condition (WHO)		
Underweight	59	5.89
Normal	762	76.12
Overweight	175	16.79
Obesity	21	2.10
Education level		
Secondary	312	31.17
University	681	68.03
Missing values	8	0.80
Occupation		
Study	937	93.60
Study and work	56	5.59
Others	8	0.80
Health history		
Diabetes	16	1.60
Arterial hypertension	11	1.10
Family health history		
Overweight or obesity	526	52.55
Diabetes	447	44.66
Arterial hypertension	406	40.56

**Table 3 behavsci-14-00154-t003:** Model fit indices.

Model	$X^{2}$	df	CFI	TLI	RMSEA	RMSEA LOW	RMSEA HIGH	SRMR
Model 9 IV and 10 DV	217.68 **	95	0.968	0.954	0.036	0.030	0.042	0.033

Note: ** = *p* < 0.01. Estimator = MLR.

**Table 4 behavsci-14-00154-t004:** Direct, indirect, and total effects of variables related to nutritional status (BMI), Low Self-Esteem, Overeating, and Lack of appetite.

Dependent Variable	Independent/Intervening Variable	Total Effect	Direct Effect	Indirect Effect
Nutritional status (BMI) r^2^ = 0.527	Self-classification by weight	0.637	0.603	0.034
	Sex	−0.499	−0.621	0.122
	Family overweight	0.441	0.219	0.222
	Appearance evaluation	−0.193	0	−0.193
	Overweight preoccupation	0.127	0.127	0
	Appearance orientation	−0.121	−0.116	−0.005
	Emotional dysregulation	0.063	0.063	0
	Interpersonal alienation	0.049	0.049	0
	Drive for Thinness	0.015	0	0.015
	Body Areas Satisfaction	−0.080	0	−0.080
	Low self-esteem (ineffectiveness)	−0.037	0	−0.037
	Somatization	0.011	0	0.011
	Anxiety	0.008	0	0.008
	Depression	−0.004	0	−0.004
	Interpersonal Sensitivity	−0.008	0	−0.008
Low self-esteem (ineffectiveness) r^2^ = 0.141	Family overweight	−0.224	−0.224	0
	Appearance Evaluation	−0.242	−0.220	−0.022
	Body Areas Satisfaction	−0.144	−0.193	0.049
	Somatization	−0.132	−0.132	0
	Overweight preoccupation	−0.122	−0.122	0
	Drive for thinness	−0.055	0	−0.055
	Self-classification by weight	−0.032	0	−0.032
	Interpersonal sensitivity	−0.012	0	−0.012
Overeating r^2^ = 0.155	Interpersonal Sensitivity	0.305	0.305	0
	Drive for thinness	−0.246	−0.246	0
	Overweight preoccupation	0.153	0.153	0
Lack of appetite r^2^ = 0.281	Overeating	−0.339	−0.339	0
	Depression	0.309	0.309	0
	Somatization	0.239	0.239	0
	Interpersonal sensitivity	−0.103	0	−0.103
	Drive for thinness	0.083	0	0.083
	Overweight preoccupation	−0.052	0	−0.052

## Data Availability

The data presented in this study are available on request from the corresponding author. The data are not publicly available due to privacy.

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
