# Peer review of "Influence of Body Image, Risk of Eating Disorder, Psychological Characteristics, and Mood-Anxious Symptoms on Overweight and Obesity in Chilean Youth"

_behavsci, 2024, doi:10.3390/bs14030154_

Round 1
Reviewer 1 Report
Comments and Suggestions for Authors
Dear Authors,
I found the topic of your research very interesting and with a good applicability.
Neverthless, in my opinion, the research cannot be accepted for publication in this form because of some methodological concerns.
In fact, adolescents are still shaping their personality traits, in particular their body image (Leggett-James, M. P., & Laursen, B. (2023). The Consequences of Social Media Use Across the Transition Into Adolescence: Body Image and Physical Activity. The Journal of Early Adolescence, 43(7), 947-964. https://doi.org/10.1177/02724316221136043).
Therefore, even though the aim of the study is very interesting and the results may lead to relevant practical applications for stakeholders and educators, the sample cannot include participants belonging to such different age ranges: 15-18 (adolescents) and 19-28 (young adults). Moreover, you used the same questionnaire for the different ages and they might not be reliable.
For instance, you stated that the SCL-30-R has been validated for UNIVERSITY STUDENTS but you also included secondary-level students; thus, is it also valid for this population?
Additionally, physical activity levels were not assessed, representing another limit. A previous study (Digennaro, S.; Iannaccone, A. Check Your Likes but Move Your Body! How the Use of Social Media Is Influencing Pre-Teens Body and the Role of Active Lifestyles. Sustainability 2023, 15, 3046. https://doi.org/10.3390/su15043046) showed that physical activity positively influences body satisfaction for both males and females.Therefore, this variable might have added a more useful insight to your research.
Conclusively, in my opinion, the research might be reconsidered for publication if the age range 15-18 is removed from the analysis and if you demonstrate that the used instruments are valid for your population.
Best regards
Author Response
Thank you very much for your comments.
Responses:
- Regarding body image in adolescents, we agree that it is still developing during this stage, which underscores the importance of evaluating its dimensions and understanding the factors that influence it. As suggested by the authors recommended by reviewer (Digennaro & Iannaccone, 2023; Leggett-James & Laursen, 2022) regarding pre-adolescents, these factors include social media exposure and physical activity, which we were unable to include in the present study, adding to the limitations of the research.
The study also had limitations in that it was not possible to assess variables that are also related to the nutritional status, body image, and mental health of adolescents, such as physical activity and the use of social media [57,58]. It would be highly beneficial to include these factors in future research, considering an external evaluation of BMI, which would allow for improved accuracy and validity of the results.
- Regarding the SCL 90-R, it should be noted that there is limited research on the psychometric properties of the instrument in Latin American adolescent populations (Londoño et al., 2019; Quiroz, 2017), and specifically in Chile, there is a lack of studies. However, the SCL 90-R is an instrument that can be applied to adolescents in the general population (Hildenbrand et al., 2015), and its utility in adolescents and young adults is emphasized:
“Indicatively, the SCL-90-R has been successfully employed for the psychopathological examination of: a) female adolescents presenting with binge eating disorder, anorexia, and bulimia nervosa and their parents (Tafà et al., 2017); b) early adolescents, who had lost a parent within their first years of life, and their surviving parents (Tafà et al., 2017)… In that context, the SCL-90-R subscales present to be of particular relevance regarding the psychopathological symptoms of adolescents and young adults (Preti et al., 2019)” (Gomez et al., 2021, p.2)
- Regarding the age range of the sample, we followed another reviewer's suggestion and removed cases over 23 years old. Although all participants were university students, we retained those within the expected age range for their studies.
- Additionally, the distributions of the SCL 90-R subscales were compared between the 15 to 18-year-old and 19 to 23-year-old groups using the Mann-Whitney U test for independent samples, with no statistically significant differences found in any of them (p>0.05).
References:
Gomez, R., Stavropoulos, V., Zarate, D., & Palikara, O. (2021). Symptom Checklist-90-Revised: A structural examination in relation to family functioning. PLoS ONE, 16(3 March). https://doi.org/10.1371/journal.pone.0247902
Hildenbrand, A. K., Nicholls, E. G., Aggarwal, R., Brody‐Bizar, E., & Daly, B. P. (2015). Symptom Checklist‐90‐Revised (SCL‐90‐R). In The Encyclopedia of Clinical Psychology (pp. 1–5). Wiley. https://doi.org/10.1002/9781118625392.wbecp495
Londoño, N. H., Agudelo, D. M., Martínez, E., Anguila, D., Aguirre, D. C., & Arias, J. F. (2019). Validación del cuestionario de 90 síntomas SCL-90-R de Derogatis en una muestra clínica colombiana. MedUNAB, 21(2), 45–59. https://doi.org/10.29375/01237047.2807
Preti, A., Carta, M. G., & Petretto, D. R. (2019). Factor structure models of the SCL-90-R: Replicability across community samples of adolescents. Psychiatry Research, 272, 491–498. https://doi.org/10.1016/j.psychres.2018.12.146
Quiroz, K. (2017). Estandarización del cuestionario SCL-90-R en adolescentes de educación básica regular de los distritos de LIMA-SUR. [Tesis de grado]. Universidad Autónoma del Perú.
Tafà, M., Cimino, S., Ballarotto, G., Bracaglia, F., Bottone, C., & Cerniglia, L. (2017). Female Adolescents with Eating Disorders, Parental Psychopathological Risk and Family Functioning. Journal of Child and Family Studies, 26(1), 28–39. https://doi.org/10.1007/s10826-016-0531-5.
Reviewer 2 Report
Comments and Suggestions for Authors
In this manuscript the authors discuss the predictor variables of nutritional status in young Chileans, focusing on various factors such as sex, family history of being overweight, self-classification by weight, body image dimensions, emotional dysregulation, and personal evaluation. My specific observations are as follows:
1. The authors must more detailed explanations for certain findings, such as the inverse relationship between Appearance Orientation and Interpersonal Alienation.
2. The authors must elaborate further on the implications of emotional dysregulation, especially considering the low coefficient obtained. It would be beneficial to the reader if the practical implications and relevance to the study's overall objectives were discussed briefly.
3. The limitations section is well-addressed, particularly in acknowledging the self-reported nature of BMI measurements. However, consider discussing the potential implications of underreporting BMI and the impact on the study's validity.
4. The statement about women experiencing higher levels of body dissatisfaction and being at greater risk for eating disorders is crucial. Thus, the authors should elaborate on why there is a gender difference.
Author Response
Thank you very much for your comments.
Responses:
- Regarding the inverse relationship between Appearance Orientation and Interpersonal Alienation, we added the relationship between depersonalization and reduced attention to body parts (La Marra et al., 2022):
(325). Along the same lines, interpersonal distrust is understood as a difficulty in ex-pressing personal thoughts and emotions, because of experiencing feelings of disappointment, disillusionment, and alienation concerning social relationships. In this regard, Calderón et al. [29] state that as body weight increases, concern for their social environment increases along with insecurity in their relationships, facilitating the rise of social problems and tendencies towards self-isolation. This relationship is explained by La Marra et al., [30] who consider that interpersonal mistrust is developed and sustained by the idealization of the culture of thinness, which generates prejudices and negative attributions towards those who are overweight or obese. This discrimination hinders psychological well-being for these people. The findings of this initial study indicate the simultaneous presence of two seemingly contradictory trends, which could actually be two aspects of the same issue. Obese adolescents expressed widespread dissatisfaction with their body image, coupled with a wish to shed weight. Conversely, they also exhibited a tendency towards depersonalization and limited self-awareness specifically linked to the most dissatisfactory body parts. (105)
This phenomenon can be analyzed considering the inverse relationship between Appearance Orientation and Interpersonal Alienation (-0.107), allowing us to hypothesize that, in the field of alienated interpersonal relationships, people would tend to avoid appearance concerns, because they would interfere with such relationships. This may relate to the findings of La Marra et al [30], in the sense that the tendency to depersonalization may decrease attention to one's own body. This aspect merits further research.
- Regarding the implications of emotional dysregulation, we detailed findings from Casagrande et al. (2020) explaining the relevance of this variable:
(334). Regarding Emotional Dysregulation, although the coefficient obtained is low (0.063), it is interesting as a variable that receives the influences of anxious (0.211) and depressive (-0.122) symptomatology, where higher amounts of anxiety would increase emotional dysregulation, and greater depressive symptomatology appears to decrease it, possibly due to the low levels of activation of depressive symptoms [15]. Moreover, according to Casagrande et al [20] overweight individuals may have more difficulties in recognizing emotions (alexithymia) and experience greater emotional dysregulation compared to those with normal weight. This emotional dysregulation could potentially pose a risk factor linked to higher BMI and a poorer prognosis for overweight and obesity issues. Conversely, an elevated BMI, along with the social and personal consequences it brings, may lead to emotional dysregulation as a result of efforts to self-regulate.
- We added potential implications of underreporting BMI and its impact on the study's validity:
(372). The limitations of this study predominantly stem from the self-reported nature of BMI measurements, which may compromise data accuracy. As noted by Allison et al. [55], it is anticipated that BMI and weight were underreported, while height was overreported in both sexes, particularly among obese individuals. This could be related to the low proportion of self-reported participants in the "obesity" category in the sample, compared to national prevalence indicators. Nevertheless, alternative research suggests that self-reporting may not be precise in predicting BMI at an individual level; however, it remains a straightforward and valid method for estimating BMI among overweight and obese individuals in epidemiological investigations [56].
The study also had limitations in that it was not possible to assess variables that are also related to the nutritional status, body image, and mental health of adolescents, such as physical activity and the use of social media [57,58]. It would be highly beneficial to include these factors in future research, considering an external evaluation of BMI, which would allow for improved accuracy and validity of the results.
- An explanation was added regarding gender differences in body image dimensions, appearance orientation, and weight self-classification:
(331). Additionally, women tend to show a stronger focus on appearance (0.245) and a higher self-classification by weight (0.309), aligning with findings from prior research [26], and which is a risk factor for ED [52] and mental health problems [53].
(396). Women tend to experience higher levels of Appearance Orientation and Self-Weight Classification than men and are at greater risk for eating disorders, placing them at risk for problems in their overall mental health.
References:
Casagrande, M., Boncompagni, I., Forte, G., Guarino, A., & Favieri, F. (2020). Emotion and overeating behavior: effects of alexithymia and emotional regulation on overweight and obesity. Eating and Weight Disorders-Studies on Anorexia, Bulimia and Obesity, 25(5), 1333–1345. https://doi.org/10.1007/s40519-019-00767-9
La Marra, M., Messina, A., Ilardi, C. R., Staiano, M., Di Maio, G., Messina, G., Polito, R., Valenzano, A., Cibelli, G., Monda, V., Chieffi, S., Iavarone, A., & Villano, I. (2022). Factorial Model of Obese Adolescents: The Role of Body Image Concerns and Selective Depersonalization—A Pilot Study. International Journal of Environmental Research and Public Health, 19(18). https://doi.org/10.3390/ijerph191811501
Reviewer 3 Report
Comments and Suggestions for Authors
An interesting piece of work that addresses a relevant issue. A good sample size and clearly expert data analysis.
However, there are a few issues that warrant consideration.
1). The argument for doing the work is based on predictors of obesity. The authors do not report the number of obese participants. They could do if they used a different approach to how they analysed the data. It would be helpful to look into the data from the intensity of data. The relationships do not untangle such information very well.
2. The sample - simple remove the older people for whom there are very few and keep data under 23.
3. Your data are not pre and post and so you have correlates. You need to explain the hypothesized model - it appears in the results but is not developed.
4. The model is almost saturated making fit likely - in the explanation of results, focus more on the strength of relationships.
5. Be more critical of the work and set up your further study which is more rigorous.
Comments on the Quality of English LanguageThe English is fine; some typos that will likely to be found as the work is edited through revisions.
Author Response
Thank you very much for your comments.
Responses:
- Regarding the obese participants, the classification of the participants' nutritional status is in Table 1.
Table 1. Sociodemographic characteristics.
Sociodemographic characteristics |
N |
% |
Total |
1,001 |
100 |
Sex Male Female |
431 592 |
43.06 56.94 |
Age (years) 15 -17 18-20 21-23 |
265 446 290 |
26.47 44.56 28.97 |
Nutritional condition (WHO) Underweight Normal Overweight Obesity |
59 762 175 21 |
5.89 76.12 16.79 2.10 |
Education level Secondary University Missing values |
312 681 8 |
31.17 68.03 0.80 |
Occupation Study Study and work Others |
937 56 8 |
93.60 5.59 0.80 |
Health history Diabetes Arterial hypertension |
16 11 |
1.60 1.10 |
Family health history Overweight or obesity Diabetes Arterial hypertension |
526 447 406 |
52.55 44.66 40.56 |
It should be noted that we preferred to use BMI as a continuous scale because it allows for more precise analyses than categories (e.g. underweight, normal, overweight, or obese).
- Participants over 23 years old were removed, and all analyses were redone.
- The explanation of how the model was constructed based on previous research was added:
(261). The hypothesized model first considered the dimensions of body image that had shown a direct relationship with BMI in previous studies (SCW, OP, AO), as well as interpersonal relationships (IA) and emotional aspects (Edy). It aimed to identify relationships between gender, age, and low self-esteem with these variables. Subsequently, depressive, and anxious symptoms related to risk behaviors (Overeating and Lack of Appetite) were included in the model, organizing the remaining variables in interrelation, based on substantive criteria, either as antecedents or covariates, obtaining a model composed of 19 variables, 9 of which are exogenous.
- Emphasis has been added on the relationships between the variables (see Results).
- Limitations and a proposal for a future study have been included:
(372). The limitations of this study predominantly stem from the self-reported nature of BMI measurements, which may compromise data accuracy. As noted by Allison et al.[55], it is anticipated that BMI and weight were underreported, while height was overreported in both sexes, particularly among obese individuals. This could be related to the low proportion of self-reported participants in the "obesity" category in the sample, compared to national prevalence indicators. Nevertheless, alternative research suggests that self-reporting may not be precise in predicting BMI at an individual level; however, it remains a straightforward and valid method for estimating BMI among overweight and obese individuals in epidemiological investigations [56].
The study also had limitations in that it was not possible to assess variables that are also related to the nutritional status, body image, and mental health of adolescents, such as physical activity and the use of social media [57,58]. It would be highly beneficial to include these factors in future research, considering an external evaluation of BMI, which would allow for improved accuracy and validity of the results.
Round 2
Reviewer 3 Report
Comments and Suggestions for Authors
The work has been revised. The authors have sided more with the favourable reviewer and addressed these points. My view is that the large sample size sways the decision. The paper is exploratory and this should be emphasised.
Comments on the Quality of English LanguageThe English is fine other than some typos and some wild paragraphs!!!
Author Response
Thank you for your dedicated review and contributions to our work. Based on your suggestions, we've included an extra point in the conclusions:
Furthermore, the complexity of the proposed model makes it necessary to replicate it in other samples in order to more precisely determine the relevance and stability of the observed relationships.
Your input has been really helpful, and we appreciate it.
Round 3
Reviewer 3 Report
Comments and Suggestions for Authors
The authors have revised the paper and other than fine editing, it could be published to let readers inform an opinion on the work.
Comments on the Quality of English LanguageThe authors should read the paper closely to ensure their message is conveyed as clearly as possible.
Author Response
Thank you so much for your helpful suggestions on our manuscript. We've requested an English reviewer to examine the language to ensure everything is in good shape. Thanks again for your time and expertise.